# The Paradox of Alcohol and Food Affordability: Minimal Impact of Leading Beer and Cachaça Brands on Brazilian Household Income Amid Hazardous Drinking Patterns

**DOI:** 10.3390/nu16101469

**Published:** 2024-05-13

**Authors:** Ian C. C. Nóbrega, Rhennan V. L. Marques, Matheus A. Ferreira, Dirk W. Lachenmeier

**Affiliations:** 1Department of Food Engineering, Universidade Federal da Paraíba (UFPB), João Pessoa 58051-900, Paraíba, Brazil; rhennan.victor@gmail.com (R.V.L.M.); mas22702@gmail.com (M.A.F.); 2Chemisches und Veterinäruntersuchungsamt (CVUA) Karlsruhe, Weissenburger Strasse 3, 76187 Karlsruhe, Germany

**Keywords:** alcohol consumption, household income, beer, dietary expenditure, Brazil, minimum unit pricing, alcohol pricing, risky drinking patterns

## Abstract

Alcohol consumption, associated with various cancers, mental disorders, and aggressive behavior, leads to three million deaths globally each year. In Brazil, the alcohol per capita consumption among drinkers aged 15 and over is 41.7 g of pure alcohol/day (~1 L beer/day), which falls into the risky consumption category and exceeds the global average by almost 30%. An effective way to mitigate alcohol-related harm is to increase its retail price. This study assesses the costs of consuming leading brands of beer and sugarcane spirit cachaça (Brazil’s most popular alcoholic beverages) against the expenditure on staple foods. Data on food and alcoholic beverage prices were collected in João Pessoa, Brazil, for 2020 and 2021. The cost per gram of pure alcohol and food were considered to establish consumption patterns of 16.8 g/day (moderate), 41.7 g/day, and 83.4 g/day (heavy), distributed in three scenarios involving the beverages alone or combined (64% beer and 36% cachaça), and a balanced 2000 kcal/day staple diet. The study finds that all heavy consumption scenarios cost less or significantly less (cachaça alone) than a 2000 kcal/day staple diet, highlighting an urgent need for fiscal policies, such as a minimum unit pricing for alcohol, to address public health concerns.

## 1. Introduction

Globally, alcohol consumption presents a significant public health challenge, with nearly half of the population aged 15 and over having never consumed alcohol, and approximately 43% identified as current drinkers. The World Health Organization (WHO) reported an average annual per capita consumption of 6.4 L of pure alcohol worldwide in 2016, a figure that significantly increases to 15.1 L when considering only those who drink [1]. A substantial portion of this consumption, up to 25.5% globally, involves unrecorded alcohol, which poses additional risks due to a lack of quality control and potential for higher alcohol content [1,2]. The WHO aims to reduce alcohol consumption by 10% by 2025, emphasizing the need for tailored national policies to address this issue.

In Brazil, alcohol consumption rates exceed the global average, with the drinking population consuming an average of 19.3 L of pure alcohol per year. This consumption pattern is predominantly driven by beer (62%), followed by spirits (34%), and wine (3%), reflecting a preference that aligns with the broader Region of the Americas [1]. The First National Survey on Alcohol Consumption Patterns in the Brazilian Population reveals that over half of Brazilians aged 18 and older consume alcohol, with significant disparities between genders and drinking frequency. The south of Brazil shows higher rates of regular drinking, contrasting with the northeast, where half of the population abstains [3].

The impact of alcohol consumption extends beyond individual health, contributing to significant morbidity and mortality worldwide [4]. In 2016, alcohol-related harm resulted in approximately three million deaths globally, surpassing deaths caused by diseases such as tuberculosis, HIV/AIDS, and diabetes. The health consequences of alcohol consumption include toxic effects on organs, alcohol dependence, and the immediate psychoactive effects of intoxication. In Brazil, alcohol consumption was responsible for a considerable number of deaths from liver cirrhosis, traffic accidents, and cancer in 2016, highlighting the urgent need for effective interventions to mitigate these harms [1].

Alcohol policies may encompass a broad spectrum of regulations that address the relationship between alcohol consumption and its implications for safety, health, and social welfare [5]. These policies extend from the production processes over the pricing strategies and availability of alcoholic beverages to bans or comprehensive restrictions on alcohol advertising.

In the context of Brazil, the enactment and implementation of alcohol policies have encountered considerable delays and exhibit substantial shortcomings. Despite the introduction of the first restrictions on hazardous substances in the Republican Code of 1890, it was not until the early 21st century that Brazil formulated and enacted concrete alcohol policies. An important moment in this trajectory was in 2005, during Brazil’s hosting of the 1st Pan-American Conference on Public Policies on Alcohol, supported by the Pan-American Health Organization. This conference catalyzed the development of intercontinental policies aimed at mitigating the adverse effects associated with alcohol consumption. It advocated for the nations in the Americas to devise strategies and programs dedicated to curtailing and preventing the harms of excessive alcohol use [6].

Following this critical juncture, the Brazilian government, in 2007, issued Decree No. 6.117 [7], sanctioning the National Alcohol Policy. This decree delineates a comprehensive strategy aimed at diminishing the misuse of alcohol and its correlation with violence and crime, alongside introducing other related measures. It embodies a concerted effort to mitigate health and life risks associated with alcohol use.

Internationally, various strategies have been deployed by governments to manage alcohol-related issues, with the regulation of pricing and taxation on alcoholic beverages being the predominant approach [8]. In this vein, Scotland’s implementation of the Minimum Unit Pricing (MUP) in 2018 stands as a significant exemplar. The MUP policy, predicated on setting a minimum price per unit of alcohol (10 mL or 8 g of pure alcohol) for retail sales, encompasses a holistic suite of policy measures and legislative initiatives aimed at reducing alcohol consumption across the population and thereby diminishing the consequent health and social detriments [9].

An investigation conducted by O’Donnel et al. [10] into the immediate repercussions of the MUP policy unveiled that the initiative successfully achieved its objectives. Notably, the most significant reductions in consumption were observed among high-alcohol-content beverages, which experienced a relative price increase compared to lower-alcohol-content alternatives. This outcome underscores the efficacy of MUP in steering consumption patterns towards safer levels, highlighting the potential of targeted policy interventions in addressing public health concerns related to alcohol consumption.

In summary, the most promising strategies for reducing alcohol-related death and damage include increasing the retail price of alcoholic beverages. Implementing such a policy in Brazil necessitates a comprehensive understanding of alcohol pricing and its impact on the nation’s average per capita income. This study aims to evaluate the effects of home consumption scenarios of beer and cachaça, the country’s two most popular alcoholic drinks, on national per capita household income. The assessment will reference the per capita expenditure on a staple balanced diet of 2000 kcal/day, providing critical insights into the economic implications of alcohol consumption patterns within Brazil.

## 2. Materials and Methods

### 2.1. Defining Levels of Alcohol Consumption

To accurately assess the impact of alcohol consumption on health outcomes, it is essential to delineate specific levels of consumption that reflect realistic quantities of alcohol intake. Following the classification system proposed by Rumgay et al. [11], alcohol consumption was categorized into three distinct levels: moderate consumption (<20 g pure ethanol/day), risky consumption (20 to 60 g/day), and heavy consumption (>60 g/day). These categorizations facilitate a nuanced analysis of alcohol’s effects across different consumption patterns.

For the purpose of this study, precise levels within these categories were established to align with the specific context of alcohol consumption in Brazil. Moderate consumption was set at 16.8 g/day, mirroring the per capita consumption of the Brazilian population over the age of 15 in 2016, as reported by the WHO [1]. Risky consumption was defined at 41.7 g/day, corresponding to the per capita consumption among Brazilian men and women over 15 who consume alcohol. Heavy consumption was determined at 83.4 g/day, a figure derived by doubling the per capita consumption indicative of risky consumption (Table 1). The selection of these specific levels was informed by the WHO’s guidelines, which acknowledge that no level of alcohol consumption is entirely devoid of health risks [12]. The chosen levels for moderate and risky drinking were based on Brazil’s alcohol per capita consumption (APC) for 2016, adjusted from liters of pure alcohol per year to grams of pure alcohol per day using a standardized formula. The heavy drinking threshold was similarly calculated, emphasizing the study’s alignment with recognized standards and the relevance of these levels to Brazil’s drinking patterns in 2016 [1].

### 2.2. Selection of Beverage Brands

The study considered the distribution of alcohol consumption by beverage type, based on publicly available WHO data from 2016 [1], with beer and cachaça identified as the primary alcoholic beverages consumed in Brazil. This distribution informed the selection of low-price leading national beer and cachaça brands for surveying in 2020 and 2021, based on data from the Brazilian Association of Supermarkets (ABRAS) [13].

The top-selling beer and cachaça brands in 2019, as identified by ABRAS [13], were selected for inclusion in the study. This selection aimed to reflect the most widely consumed and accessible brands in Brazil, facilitating a realistic assessment of alcohol purchase and consumption patterns. The number of beer and cachaça samples corresponded to the availability of selected brands in visited supermarkets/grocery stores, ensuring a representative collection of data on popular alcoholic beverages. Regular prices for the selected beer and cachaça brands were collected from local supermarkets or grocery stores, excluding promotional or limited-time discount offers to maintain consistency in price analysis.

The average price per gram of pure alcohol was determined by dividing the average product price (in Brazilian reals, BRL) by the mass of ethanol in the beverage, using the density of ethanol (0.79 g/mL) as a conversion factor. This calculation provides a basis for comparing the cost-effectiveness of different alcoholic beverages in relation to their alcohol content, offering insights into consumption preferences based on price and alcohol content.

### 2.3. Choice of Alcoholic Beverages and Definition of Consumption Scenarios

To accurately assess the expenditure on alcohol at the defined daily consumption levels, the selection of alcoholic beverages for price investigation in the retail market was guided by prevalent consumption patterns in Brazil. Utilizing the WHO’s 2018 distribution data on per capita alcohol consumption in Brazil for individuals over 15 years old, the following beverage types were identified: beer (61.8%), spirits (34.3%), wine (3.4%), and other alcoholic beverages (0.5%) (Table 2). Given the lack of specificity in the WHO (2018) report regarding the types of spirits consumed, further refinement was necessary for this study.

The research by Laranjeira et al. [3], which examines alcohol consumption patterns within the Brazilian population, indicated cachaça as the predominant spirit, accounting for approximately 42% of spirit consumption. Vodka followed with an 18% consumption rate. Based on these insights, and to streamline the research process, this study focused exclusively on the price analysis of selected brands of cachaça and beer. This decision reflects both the prominence of these beverages in the Brazilian market and their relevance to the study’s objectives of determining alcohol expenditure at specified consumption levels.

The choice to limit the study to these two types of beverages is aimed at ensuring both the manageability of the research process and the relevance of the findings to the broader context of alcohol consumption in Brazil. This approach facilitates a focused examination of spending behaviors associated with the most popular alcoholic beverages, thereby enabling the derivation of insights that are both meaningful and actionable within the context of Brazilian alcohol consumption trends.

### 2.4. Choosing Food and Defining a Daily Diet

The selection of food staples and the definition of a 2000 kcal/day diet for this study were planned to reflect the dietary habits prevalent in Brazil as a whole; for this purpose, aspects of Brazilian dietary patterns and healthy meal options, as described by the dietary guidelines for the Brazilian population [14], were considered (Table 3). This entailed surveying local supermarkets/grocery stores to identify the reference weight (kg) or volume (mL) for the sale of essential food items, alongside determining their corresponding values for edible conversion factors (ECFs), cooking conversion factors (CCFs), and total conversion factor (TCF). Additionally, the energy content (kcal per 100 g or 100 mL) and selected nutrient content (g or mg per 100 g or 100 mL) of these foods in their usual form of consumption were assessed based on the Brazilian Food Composition Table [15].

The basic food staples surveyed included whole/fresh banana, rice, beans, tomato, beef chuck, and other staples integral to the Brazilian diet. The average weights for these items were sourced from various studies and reports, such as Almeida et al. [19] for pineapple and CEAGESP [20] for lettuce (see also the details in Appendix A, Table A1).

The usual form of food consumption was aligned with known dietary habits in Brazil, incorporating preparation methods that are commonplace in Brazilian cuisine, such as steaming for corn couscous and sautéing for white rice. Serving sizes for each food item, such as soybean oil, fluid milk, raw banana, and cooked rice, were determined based on estimations from authoritative sources like ANVISA and the Ministry of Health [16,21].

ECFs, CCFs, and TCF for the surveyed food commodities were calculated to convert the weight or volume of food as sold into its edible form post-preparation. For instance, the TCF calculation for raw beef chuck eye steak involved dividing the CCF by the ECF, providing a quantifiable measure of the food in its final form of consumption. The calorie and nutrient content of each food item, including trace levels, were sourced from TBCA [15], ensuring the accurate representation of their nutritional value in the typical Brazilian diet. Levels and recommendations for selected dietary components for adults were referenced from various health guidelines, including the Agência Nacional de Vigilância Sanitária (ANVISA) and the World Health Organization (WHO) [16,18]. These references guided the formulation of a balanced and adequate diet from the surveyed food staples, adhering to the latest dietary guidelines and nutrient requirements.

### 2.5. Place and Period of Price Collection for Alcoholic Beverages and Foodstuffs

The data collection for the prices of alcoholic beverages and foodstuffs was conducted in João Pessoa, the capital city of the northeastern state of Paraíba, Brazil. This region, with a population of approximately 800,000, served as the focal area for this study. The period designated for price collection spanned two phases: between August and September of both 2020 and 2021, allowing for an assessment of price variations over time within the context of the COVID-19 pandemic.

To ensure a comprehensive and representative dataset, the selection process for supermarkets and grocery stores was comprehensively planned. Initial research was conducted through internet searches, supplemented by site visits, to identify potential establishments for data collection. These establishments were located within a circular geographic area with a radius of 15 km, centered around the Universidade Federal da Paraíba (UFPB), encompassing an area of approximately 700 km^2^. This strategy was aimed at capturing a wide range of retail environments within the city, from large supermarkets to smaller grocery stores, to accurately reflect the retail landscape of João Pessoa.

Given the unique challenges posed by the COVID-19 pandemic, specific health and safety measures were rigorously observed during all visits to supermarkets and grocery stores. Researchers equipped themselves with N-95 disposable respirators to mitigate the risk of virus transmission. In adherence to guidelines issued by health authorities, additional precautions included maintaining increased space and practicing distancing from other individuals during site visits. These measures ensured the safety of both the researchers and the public, allowing for the successful collection of price data amidst the pandemic.

### 2.6. Determining the Final Price of Meals

To accurately assess the economic implications of dietary choices, the final price of meals, incorporating both branded and non-branded food items, was calculated. For non-branded food items such as oranges, onions, and raw/chilled beef chuck, the regular price recorded was the one available at each visited supermarket or grocery store. This approach captures the typical market price for staple food items that do not vary significantly by brand. For branded food items like soybean oil, rice, and ground coffee, where multiple brands were often available, the lowest regular price was consistently selected. This decision was made to reflect cost-effective purchasing habits, excluding “best buys” or temporary discount offers to maintain price consistency across data collection periods.

The number of food staple samples correlates directly with the number of supermarkets and grocery stores visited where the specific item was available, ensuring a broad and representative dataset for price analysis. To derive the average price of food per reference weight or volume as sold, the calculated cost per gram or milliliter of the converted food is adjusted by the TCF and the reference sale weight or volume.

This structured methodology for calculating meal prices allows for a comprehensive analysis of dietary economics, factoring in the variations in food preparation and consumption practices. It provides a detailed basis for evaluating the cost implications of dietary choices within the context of Brazilian households.

### 2.7. Assessing the Impact of Alcohol and Food Consumption on National Per Capita Household Income

The economic burden of alcohol and food consumption on Brazilian households was analyzed by comparing these expenditures to the national per capita household income. Such an analysis reveals the percentage of income that households dedicate to alcohol and food, offering insights into the economic pressures they face.

National per capita household income data were sourced from the Brazilian Institute of Geography and Statistics (IBGE), Ministry of Economy. According to the IBGE, the nominal monthly per capita household income of the resident population in Brazil was BRL 1380.00 in 2020 and slightly decreased to BRL 1367.00 in 2021 [22,23]. These figures provide a basis for evaluating the relative economic impact of alcohol and food expenditures on household budgets. The expenditure on alcohol and food was calculated using price data collected from supermarkets and grocery stores in João Pessoa, as specified above. The total expenditure on alcohol and food for an average household was then calculated. This amount was divided by the national per capita household income to determine the expenditure ratio. This ratio, expressed as a percentage, illustrates the share of household income consumed by these expenditures.

## 3. Results

### 3.1. Average Price of Pure Alcohol from Leading Brands of Beer and Cachaça

Selected leading national brands of beer and sugarcane spirit cachaça surveyed in local (João Pessoa, Brazil) supermarkets/grocery stores in 2020 and 2021, and corresponding average prices (per product and per gram of pure alcohol), are shown in Table 4.

Price variations of the branded beverages in different supermarkets/grocery stores surveyed in 2020 and 2021 were within expectations. The coefficient of variation (CV, standard deviation divided by the average and multiplied by 100) of all brands ranged from 7.7% (brand B1 in 2020) to 13.2% (brand C1 in 2021), with an overall CV of 10.2%.

From 2020 to 2021, the average price of the beer and cachaça brands rose by 11.9% and 9.6%, respectively, and these increases are aligned with the official annual inflation rate of 10.6% in 2021 (in 2020, the inflation was 4.52%) [24,25].

On average, the price per gram of pure alcohol from cachaça is approximately four times cheaper than that from beer, both in 2020 and 2021 (Table 4). This difference in price may be related to the fact that beer produced in Brazil depends heavily on imported raw materials (especially malt and hops), which are more subjected to price variations in the international market, a fact that does not apply to cachaça (Brazil is the world’s largest producer of sugarcane). Despite the peculiarities in production costs between beer and cachaça, such an impressive discrepancy in price per gram of pure alcohol was unexpected.

As the significant difference in average price of pure alcohol from brands of cachaça (BRL 0.03 and BRL 0.04) and beer (BRL 0.12 and BRL 0.17) could have been the result of specific market disruptions during the COVID-19 pandemic in 2020 and 2021, in February 2024, while this article was being written, a representative supermarket chain in João Pessoa, Paraíba, Brazil, was visited to check how different the prices were from the 2021 survey. Brands B1, B2, B3, C1, and C2 were available in the visited supermarket, all in the same type of packaging material/volume (aluminum cans of 350 mL). After more than two years since the last survey, both the B1 (5.0% vol.) and B2 (4.7% vol.) beer brands were being sold at BRL 3.19 while the price of B3 (4.5% vol.) was BRL 2.87. Regarding the cachaça brands, prices of both C1 (39% vol.) and C2 (40% vol.) were BRL 4.99. Assuming this single price collection in 2024 represents the current average prices in the Brazilian market, it appears that while prices of the leading beer brands rose sharply since 2021, with an average increase of 38.6%, the price of cachaça rose 12.4%. After applying the same conversion procedure indicated in Table 4, prices per gram of pure alcohol in February 2024 were 0.23 (B1), 0.25 (B2), 0.23 (B3), 0.05 (C1), and 0.04 (C2); in this respect, the average gram of pure alcohol from cachaça (BRL 0.045) would be five times cheaper than that from beer (BRL 0.24) in 2024.

It should be highlighted that the leading brands of cachaça surveyed in 2020 and 2021, which are known to be low-price, were not the cheapest brand available in some supermarkets. For example, a 38% vol. recorded brand of cachaça that was being sold at BRL 2.69 in a PET bottle contained 480 mL, which converts into BRL 0.02 per gram of pure alcohol. Furthermore, in the 2020 and 2021 survey, some brands of vodka, the second most consumed spirit in Brazil [3], which is usually produced from sugarcane [26], have prices that are similar to the leading brands of cachaça.

### 3.2. Average Price Per Gram of Converted Food Staples

Table 5 presents a summary of average prices (BRL) per gram of 24 converted food staples (i.e., food without inedible/unwanted parts and then cooked, if applicable) collected from supermarkets and grocery stores in 2020 and 2021. For detailed information on food characteristics/composition, forms of consumption, forms of preparations, standard serving size, edible conversion factor, cooking conversion factor, etc., see Appendix A Table A1.

Food price variations in different supermarkets/grocery stores surveyed in 2020 and 2021 were greater than values observed for the leading brands of beer and cachaça (9.6% to 11.9%). Price variations in food, given as a coefficient of variation (CV), ranged from 3.9% (mozzarella-type cheese in 2020) to 58.3% (raw tomato cv Italiano in 2021), with a 16.7% average CV of all surveyed foods. Compared to the three brands of beer and two brands of cachaça, the greater variation in food prices in different outlets may simply reflect a more complex nature associated with the production and logistics of 24 different types of foods.

Regarding variations in average prices of the 24 food staples from 2020 to 2021, the majority (21) experienced increases while prices fell for only 2 of them (raw garlic, −20.8%; salt, −31.2%). In one case, beans cv Carioca, the average price remained the same in 2021. Foods that experienced increases in prices ranged from 2.3% (UHT whole fluid milk) to 70.8% (raw butterhead lettuce), with an overall increase in food of 30.9%. As the official inflation rate in Brazil was reported as 10.06% in 2021 [24], price increases in most surveyed food staples were well above the annual inflation rate and, again, they may have been compounded by market disruptions due to the COVID-19 pandemic.

A quick price update in February 2024 in a representative supermarket in João Pessoa, Paraíba, Brazil, was conducted to check how different the prices of 10 selected food staples were from the 2021 survey. Similar to the 2020 and 2021 surveys, the lowest price available for each item was collected. After applying the conversion factors when necessary, prices per gram or mL of the 10 selected food items in February 2024 were the following (price increase relative to the 2021 survey in brackets): banana, BRL 7.4/g (+32.1%); papaya, BRL 9.8/g (+84.9%); tomato, BRL 6.7/g (+85.8%); lettuce, BRL 17.6/g (+114.1%); milk, BRL 5.3/mL (+17.6%); rice, BRL 3.2/g (+59.0%); beans, BRL 2.7/g (+6.4%); sugar, BRL 4.2/g (+12.3%); chicken, BRL 64.6/g (+12.3%); and beef, BRL 49.3/g (+8.3%).

Assuming the single price collection in 2024 represents the current average prices in the Brazilian market, it appears that food prices have been rising at different paces since 2021. In some cases, prices rose sharply for some foods, such as lettuce, papaya, tomato, and rice (between 59 and 114%); in other cases, such as beef, chicken, sugar, and beans, a more discrete price increase (between 6 and 13%) was observed.

### 3.3. Daily Expenditures on Alcohol and Food Consumption

Table 6 and Table 7 present daily expenditures on alcohol and food consumption, respectively, both in 2020 and 2021. Table 6 shows spending in nine specific consumption scenarios involving different types of drinks (beer and/or cachaça) and the level of consumption (moderate, risky, and heavy). Table 7, on the other hand, presents the expenditure on a low-price balanced staple diet of 2000 kcal that includes 24 different types of food. The daily expenditures on alcohol and food were obtained by multiplying the daily per capita consumption of pure alcohol (in grams) and converted food (in grams or mL) by the average price of each item, which have already been specified in Section 3.1 and Section 3.2 (Table 4 and Table 5).

With respect to price variations that occurred from 2020 to 2021, expenses rose when the consumed beverages were beer alone and beer + cachaça by approximately 13% and 12%, respectively; however, there was no change when the beverage consumed was cachaça only. Compared to alcohol consumption, there was a much higher increase in food (staple diet of 2000 kcal/day) spending in 2021, around 25%. In brief, spending on alcohol consumption either did not rise or rose much less sharply than spending on food in 2021. Based on the recent single survey on the prices of alcoholic beverages (beer and cachaça) and selected foods (see Section 3.1 and Section 3.2), it is assumed that the higher rate of increase in food spending (in relation to alcohol) remains or even increased in 2024.

All expenses related to the nine alcohol consumption scenarios in 2020 and 2021 (Table 6), which varied from BRL 0.67/day (moderate consumption of cachaça, equivalent to 0.4 L/week) up to BRL 13.34/day (heavy beer consumption, equivalent to 15.7 L/week), cost less than the expenditure on a staple balanced diet of 2000 kcal (BRL 14.73/day). It should be noted that all scenarios of heavy alcohol consumption are cheaper than the staple diet; in this sense, it is concerning that the scenario involving heavy daily consumption of cachaça (equivalent to 1.9 L/week) has an average cost of only one-fifth (or 20%) of the amount spent on the staple diet.

Although the beers and cachaças surveyed are considered leading brands, therefore “popular alcoholic beverages” in Brazil, when looking at cachaça and beer drunk separately, the cost of drinking the same amount of alcohol from cachaça is approximately four times lower than that from beer. As mentioned previously, a possible explanation for this discrepancy is the fact that the main raw material for manufacturing cachaça—sugar cane—is very abundant and cheap in Brazil (compared to raw materials used in breweries, especially imported malt and hops).

Although a scenario of moderate consumption was tested (Table 6), it is worth noticing that the latest epidemiological evidence indicates that there is no safe amount of alcohol that can be consumed (to the point of excluding risks associated with alcohol completely), as alcohol is toxic to the human body at any level; therefore, the level of alcohol consumption that minimizes health loss is zero [33]. However, moderate levels of alcohol consumption have been defined, such as less than 20 g of pure alcohol per day [11].

### 3.4. Impact of Daily Expenditures in Alcohol and Food Consumption on National Per Capita Household Income

The question that arises is to what extent food and alcohol spending scenarios impact on the finances of the average Brazilian. In other words, how much of the national income does food and alcohol consumption represent? It is currently understood that alcohol consumption, in any of the presented scenarios, represents a smaller portion of a Brazilian’s income compared to a 2000 kcal staple diet.

The first aspect to consider is that food spending is necessary for survival and, therefore, mandatory. Let us consider, then, that the expenditure on the balanced staple diet of 2000 kcal would already be at the minimum possible cost (average of BRL 14.73/day) and, therefore, with no more room for reductions. In this case, one can ask the following: would there be, in theory, room in the household income to allow for the inclusion of the alcohol consumption scenarios, in particular the risky and heavy scenarios, without compromising other mandatory expenses (housing, transport, health, clothing, etc.)?

According to the Brazilian Institute of Geography and Statistics—IBGE—the monthly national household income per capita in Brazil in 2020 and 2021 was BRL 1380.00 and BRL 1367.00, respectively. Per capita household income is calculated as the ratio between the total household income and the total number of residents [22,23]. Considering the 2020–2021 income average and dividing the value by 30, the daily household income in those years was BRL 45.78.

Figure 1 shows the impact (%) of daily alcohol and food consumption on the daily national per capita household income. They were obtained by dividing average daily expenditure values (2020 and 2021) on food and alcohol (Table 6 and Table 7) by BRL 45.78 and multiplying by 100.

The balanced diet of 2000 kcal/day (BRL 14.7 3/day, average over 2020 and 2021) represents a significant portion of household income, 32%, the biggest impact of all the scenarios analyzed (Figure 1). The variations in the impact of daily expenditures on national per capita household income between 2020 and 2021, calculated as a coefficient of variation (CV), ranged from 0.0% (all cachaça scenarios) to 11.0% (diet), with an overall value of 4.6%.

Taking into account the moderate scenario (16.8 g of pure alcohol per day) only, the consumption of cachaça alone, cachaça + beer, or beer alone has very little impact on household income, namely 1.5%, 4.3%, and 5.9%, respectively. If the consumption moves to a risky (41.7 g/day) scenario, which corresponds to the per capita consumption among Brazilian men and women over 15 who consume alcohol, impacts on the domestic income are 3.6% (cachaça alone), 10.6% (beer + cachaça), and 14.6% (beer alone) (Figure 1).

Finally, in a scenario of heavy alcohol consumption (83.4 g/day), impacts on household income are 7.3% (cachaça alone), 21.3% (beer + cachaça), and 29.1% (beer alone), all below the expenditure on a 2000 kcal staple diet. Although the volumes of beverage consumption in the heavy scenario are all very high, it is the cachaça scenario and its impact on household income that draws the attention most: consumption of a large amount of cachaça (equivalent to 1.9 L per week) has minimal impact, 7.3%, on the household income (Figure 1). From the financial point of view, the result suggests that it is entirely possible for average Brazilians to engage in a heavy alcohol consumption pattern, especially from drinking leading brands of cachaça, without compromising other mandatory household expenses. Engaging in a risky scenario of drinking is thus even more economically viable.

## 4. Discussion

The health and economic burdens associated with alcohol consumption disproportionately affect various socioeconomic groups, with the most economically disadvantaged and less educated groups experiencing more significant health impacts and financial strain. This strain includes compromises in family income and the ability to afford essential dietary needs [1]. The World Health Organization advocates for increasing the retail price of alcohol as an effective measure to mitigate the harm caused by its consumption [34]. This approach has proven successful in reducing alcohol-related harm, morbidity, and mortality when properly implemented [1]. However, the application of such a policy in Brazil necessitates a thorough understanding of alcohol’s relative pricing and its impact on household income. To date, there has been limited research in this area.

The study by Abramson et al. [35] stands out as a pioneering effort to explore this issue within Brazil. Their research involved a comparison of the prices of popular alcoholic beverages (such as cachaça, beer, and table wine) with basic food groups (cheese, milk, rice, vegetables, and meat), based on data collected from 32 retail supermarkets across eight municipalities in Paraíba. Their findings highlighted a discrepancy in pricing, with alcoholic beverages, particularly popular brands of cachaça and beer, being significantly cheaper than basic food items. This disparity suggests that interventions aimed at addressing alcohol dependence must consider the economic accessibility of alcohol relative to essential dietary foods. While Abramson et al. [35] offered crucial insights, they did not account for the typical expenditure on alcohol consumption (e.g., grams of pure alcohol per day) or compare it against the expenditure on a reference diet (e.g., 2000 kcal/day). Despite this limitation, their conclusion that consuming popular alcoholic beverages is more affordable than adhering to dietary staples remains a critical observation, underscoring the need for policy interventions that consider both the economic and health impacts of alcohol consumption within Brazil.

In relation to the per capita household income in 2020 and 2021, both approximately BRL 46.00 per day [22,23], the daily household expenditure on a 2000 kcal diet in 2020 (BRL 13.11) and 2021 (BRL 16.36) represents 28% and 36%, respectively. It is assumed that this increase is related to the official inflation measured in Brazil in those years, which jumped from 4.52% in 2020 to 10.06% in 2021 [24,25].

In addition to eating, the population must also direct their income to meet all other basic needs, such as transport, health, clothing, and housing. Therefore, committing between 28% and 36% of income to food alone is concerning. This monthly commitment percentage of 28–36% on a diet of 2000 kcal/day is in line with the latest Family Budget Survey (POF) 2017–2018 by IBGE, according to which families with up to two minimum wages a month in Brazil (in 2018, BRL 63.60/day) spent 22% on food [36].

This research approach faces some limitations that must be acknowledged. The study’s focus on beer and cachaça, while providing valuable insights into these popular beverages, may overlook the diversity of alcohol consumption patterns across different segments of the Brazilian population. This could limit the generalizability of the findings to the broader context of alcohol use in Brazil. Furthermore, the reliance on retail prices from supermarkets in Greater João Pessoa may not accurately reflect price variations and consumption patterns in other regions, particularly in rural areas or among different socioeconomic groups, as well as the phenomenon of unrecorded alcohol, which might be prevalent specifically for sugarcane spirits. This geographic limitation suggests the need for further research encompassing a broader array of locations and demographic profiles to capture the full spectrum of alcohol’s economic impact on Brazilian households.

Another limitation stems from the study’s methodological focus on economic impacts, which may not fully account for the complex social and cultural factors influencing alcohol consumption in Brazil [37]. While economic measures like Minimum Unit Pricing have the potential to mitigate harmful drinking patterns, their effectiveness can be influenced by cultural attitudes toward alcohol, social drinking norms, and the availability of informal alcohol markets. For instance, in Brazil, social gatherings and celebrations often center around alcohol, potentially reinforcing and normalizing high consumption rates. Moreover, the stigma associated with seeking help for alcohol dependency varies culturally, which can affect the effectiveness of public health interventions. Moreover, dietary habits may be intrinsically linked to alcohol consumption. In many cultures, including Brazil, alcohol is often consumed in conjunction with meals, and the types of food available or preferred can affect the amount and type of alcohol consumed. For instance, the prevalence of heavy fat- and carbohydrate-rich foods may traditionally correlate with the preference for spirits like cachaça. Understanding these dietary patterns can provide insights into the contextual factors that drive alcohol consumption. Moreover, dietary habits can also reflect broader socioeconomic conditions that influence drinking behaviors, such as income levels and access to food diversity. Addressing these multifaceted drivers of alcohol use requires a comprehensive approach that combines economic interventions with social and cultural strategies.

This article contributes to a deeper understanding of alcohol consumption’s economic impacts on Brazilian households. By focusing on beer and cachaça, the most popular alcoholic beverages in Brazil, the study provides targeted insights into consumption patterns that have significant implications for public health and economic well-being. This specificity allows for a nuanced analysis of how these beverages contribute to household expenditures and potentially displace essential dietary needs. Additionally, using the per capita expenditure on a balanced diet as a reference point offers a tangible measure of the trade-offs households may face between alcohol consumption and nutritional requirements, highlighting the broader societal impacts of alcohol affordability.

## 5. Conclusions

This study highlights a significant public health and economic concern due to the affordability of alcoholic beverages relative to essential dietary needs in Brazil. Notably, alcoholic drinks such as cachaça are often more affordable than staple foods, which may facilitate hazardous drinking patterns within average Brazilian households. This finding indicates that a reevaluation of alcohol pricing policies, including the consideration of Minimum Unit Pricing (MUP), could be beneficial. However, the actual effectiveness of such measures requires careful consideration and further empirical validation. Combining educational campaigns with strategic pricing could also offer a more comprehensive approach to mitigating alcohol-related harm. Developing effective policy responses will necessitate a multifaceted strategy that encompasses regulatory, educational, and economic measures, tailored to the complex interplay between alcohol affordability, consumption, and public health. It is essential for future research to explore the long-term impacts of these policies on alcohol consumption patterns, health outcomes, and the economic burden on households to ensure that policies are both effective and equitable.

## Figures and Tables

**Figure 1 nutrients-16-01469-f001:**
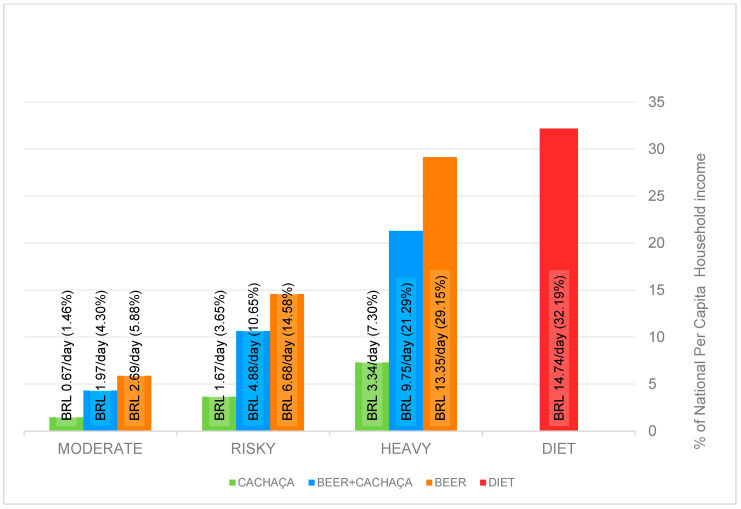
Impact (%) of alcohol and food expenditures on national per capita household income, average 2020 and 2021.

**Table 1 nutrients-16-01469-t001:** Categories of alcohol per capita consumption (APC) by range (in grams of pure alcohol per day) ^1^, APC levels considered in the study, and their relevance to Brazil.

Categories (Range) ^2^	Considered Level of APC	Relevance of the APC to Brazil ^5^
Moderate drinking (<20 g per day)	16.8 g per day ^3^	General population (15+) consumption, 2016
Risky drinking (20–60 g per day)	41.7 g per day ^3^	Drinkers only (15+) consumption, 2016
Heavy drinking (>60 g per day)	83.4 g per day ^4^	-

^1^ Although categories of the APC by range are accepted as references for assessing consumption patterns of populations and individuals, current WHO guidelines state that there is no safe level of alcohol consumption that does not affect human health [12]. ^2^ Categories, by range of grams of pure alcohol consumption per day, according to Rumgay et al. [11]. ^3^ The levels for moderate and risky drinking (in g of pure alcohol per day) were chosen because they coincide with the APC (15+, both sexes) in Brazil in 2016 [1], namely 7.8 L of pure alcohol per year (general population) and 19.3 L of pure alcohol per year (drinkers-only population); the APC in 2016 was converted from liters of pure alcohol/year to g of pure alcohol/day using the following formula: (APC in liters of pure alcohol per year × 1000 × 0.79)/366 days. ^4^ The APC considered for heavy drinking was obtained multiplying the APC for risky consumption by 2. ^5^ Relevance of chosen levels considering WHO GISAH data from Brazil in 2016 [1].

**Table 2 nutrients-16-01469-t002:** Scenarios of alcohol per capita consumption (15+, both sexes) in Brazil for moderate (16.8 g pure alcohol per day), risky (41.7 g pure alcohol per day), and heavy (83.4 g pure alcohol per day) drinking, considering the consumption of beer only, cachaça only, or a combination of the two beverages (64% beer + 36% cachaça).

Type of Alcoholic Beverage	Category of Alcohol Consumption	Consumption in g of Pure Alcohol/Day ^1^	Consumption in mL of Beverage/Week ^2^
Beer only	Moderate	16.8	3167.2
	Risky	41.7	7861.6
	Heavy	83.4	15,723.1
Cachaça only	Moderate	16.8	376.9
	Risky	41.7	935.4
	Heavy	83.4	1870.9
Combination of beverages (64% + 36%) ^3^	Moderate	10.8 + 6.0	2036.1 + 134.6
	Risky	26.7 + 15.0	5033.7 + 336.5
	Heavy	53.4 + 30.0	10,067.3 + 673.0

^1^ Values were obtained according to Table 1. ^2^ The corresponding volume (mL) of the beverages (beer or cachaça) was calculated as follows: [(consumption in g of pure alcohol per day)/0.79] × (1/alcohol content of each beverage in % vol) × 7. The alcohol contents of beer and cachaça used were 4.7% and 39.5%, respectively, because they correspond to the average alcohol contents displayed on labels of beers and cachaças selected for the present study. The density of alcohol is 0.79 g/mL. ^3^ Recorded alcohol per capita consumption (15+; in liters of pure alcohol) values by type of alcoholic beverage in 2016 according to WHO GISAH [1] were the following: 61.8% beer, 34.3% spirits, 3.4% wine, and 0.5% other beverages. Distribution in the spirits category is not provided by WHO GISAH [1], but according to a large Brazilian survey [3], cachaça is by far the most consumed spirit in Brazil. The rationale behind the combined consumption of beer and cachaça considered, first, cachaça as the only spirit consumed in the WHO GISAH spirit category; then, the consumptions of wine and other beverages (3.9%) were distributed equally to beer (61.8%) and spirits (34.3%), resulting in the 64% beer + 36% cachaça scenario.

**Table 3 nutrients-16-01469-t003:** Levels and recommendations of selected dietary components for adults (both sexes) used as references in the preparation of a balanced/adequate diet from surveyed food staples.

Component	Level/Recommendation	Source
Energy	2000 kcal/day	Health Regulatory Agency, Brazil [16]
Carbohydrates (sugars and starches)	45–65% energy	Institute of Medicine, USA [17]
Free sugars	<10% energy	Institute of Medicine [17] and WHO [18]
Protein	10–35% energy	Institute of Medicine, USA [17]
Total fat	<30% energy	WHO [18]
Saturated fat	<10% energy	WHO [18]
Trans fat	<1% energy	WHO [18]
Sodium	<2000 mg/day	WHO [18]
Fruits and vegetables ^1^	≥400 g/day	WHO [18]
Fiber	≥25 g/day	Health Regulatory Agency, Brazil [16]
Natural or minimally processed foods	Basis of diet	Dietary Guidelines, Brazil [14]
Processed foods	Limit	Dietary Guidelines, Brazil [14]
Ultra-processed foods	Avoid	Dietary Guidelines, Brazil [14]

^1^ Excluding potatoes and starchy roots, according to World Health Organization [18].

**Table 4 nutrients-16-01469-t004:** Selected low-price leading national brands of alcoholic beverages (beer and cachaça) ^1^ surveyed in local (João Pessoa, Brazil) supermarkets/grocery stores in August–September/2020 and August–September/2021, number of samples, average price (Brazilian reals, BRL) of the alcoholic beverage and average price of corresponding g of pure alcohol.

Alcoholic Beverages ^1^ and Alcohol Content (Labels), as Sold in Local Supermarkets	Number of Samples ^2^		Average Price (BRL) of Branded Product ^3^ ± Std. Deviation		Average Price (BRL) of g Pure Alcohol ^4^	
	2020	2021	2020	2021	2020	2021
Beer, can, 350 mL						
Brand B1 (5.0% vol.)	14	24	1.95 ± 0.15	2.09 ± 0.17	0.11	0.15
Brand B2 (4.7% vol.)	13	26	1.97 ± 0.22	2.35 ± 0.19	0.13	0.18
Brand B3 (4.5% vol.)	10	19	2.06 ± 0.16	2.25 ± 0.23	0.12	0.18
Average price (B1–B3 brands)			1.99	2.23	0.12	0.17
Cachaça, can, 350 mL						
Brand C1 (39% vol.)	9	17	3.84 ± 0.49	4.39 ± 0.58	0.03	0.04
Brand C2 (40% vol.)	15	14	4.28 ± 0.56	4.49 ± 0.48	0.03	0.04
Average price (C1 and C2 brands)			4.06	4.44	0.03	0.04

^1^ To define low-price leading national beer and cachaça brands to be surveyed in 2020 and 2021, data from the Brazilian Association of Supermarkets regarding leading (best-selling) brands in the year 2019 were used [13]. The 5 best-selling beer brands (brand name/manufacturer) in 2019, all manufactured in Brazil, were Skol pilsen/AMBEV, Brahma chopp/AMBEV, Budweiser/AMBEV, Heineken/Heineken Brasil, and Itaipava Pilsen/Grupo Petrópolis. The 5 best-selling cachaça brands (brand name/manufacturer) in 2019, all manufactured in Brazil, were Pirassununga 51/Müller, Velho Barreiro/Tatuzinho, Ypióca/Diageo, Pitú/Engarrafamento Pitú, and Salinas/Lactalis. Due to greater availability in local supermarkets and/or relatively lower regular prices (among the top 5 best-selling brands), 3 brands of beer (named in this study as B1, B2, and B3; all “American Lager” in beer style) and 2 brands of cachaça (named in this study as C1 and C2; all colorless/“white” sweetened column still type cachaças) were chosen to be surveyed. ^2^ The number of samples of beer and cachaça corresponds to the number of visited supermarkets/grocery stores where the specific brands of beer and cachaça (in aluminum cans of 350 mL) were available. ^3^ Regular price of product collected in local supermarkets or grocery stores (“best buys” or limited-time discount offers were not considered). ^4^ Average price per g of pure alcohol was calculated by dividing the average price (BRL) of the product by the mass of ethanol in the beverage (volume of alcoholic beverage × alcohol content in % vol. × 0.79); 0.79 g/mL is the density of ethanol.

**Table 5 nutrients-16-01469-t005:** Selected low-price ^1^ food staples available in local (João Pessoa, PB, Brazil) supermarkets/grocery stores in August–September/2020 (2020) and August–September/2021 (2021), number of samples, and average price per g or mL of converted food.

Food Staples ^2^, as Sold in Local Supermarkets/Grocery Stores	Number of Samples ^3^		Form of Consumption/ Use of Edible Portion ^2^	Average Price (BRL) of Converted Food ^4^ per g or mL ± SD (×10^3^)	
	2020	2021		2020	2021
Orange cv. Pera, raw/natural	12	26	Squeezed	4.1 ± 0.5	4.9 ± 1.3
Banana cv. Pacovã, raw/natural	12	29	Sliced	3.5 ± 0.8	5.6 ± 1.4
Papaya cv. Formosa, raw/natural	12	28	Sliced	3.5 ± 0.8	5.3 ± 1.6
Pineapple cv. Pérola, raw/natural	17	23	Sliced	5.1 ± 1.0	5.4 ± 1.3
White onion, raw/natural	13	28	As ingredient	3.1 ± 0.6	4.2 ± 1.0
Carrot, raw/natural	13	28	Grated	3.8 ± 0.5	4.7 ± 0.9
Garlic, raw/natural	19	26	As ingredient	30.7 ± 5.4	24.3 ± 3.9
Tomato cv. Italiano, raw/natural	12	26	Sliced	2.9 ± 0.3	3.6 ± 2.1
Green butterhead lettuce, raw/natural	11	26	Sliced	4.8 ± 1.5	8.2 ± 2.7
Flaked cornmeal, pre-cooked/dry, branded	13	30	Steamed	1.5 ± 0.1	1.7 ± 0.2
White/polished rice, dry/raw, branded	22	30	Cooked	1.9 ± 0.2	2.0 ± 0.3
Beans cv. Carioca, dry/raw, branded	16	29	Cooked	2.5 ± 0.2	2.5 ± 0.4
Spaghetti-type wheat pasta, dry/raw, branded	23	29	Cooked	1.4 ± 0.1	1.8 ± 0.3
Granulated cane sugar, white, branded	20	26	As sweetener	2.5 ± 0.1	3.7 ± 0.3
Table salt, refined, branded	13	26	As ingredient	1.6 ± 0.2	1.1 ± 0.3
Soybean oil, refined, branded	12	30	As ingredient	6.2 ± 1.9	9.6 ± 0.6
Mozzarella-type cheese, cow’s, branded	9	38	Sliced	36.2 ± 1.4	37.2 ± 6.0
Butter, unsalted, branded	8	14	As spread	50.3 ± 6.3	51.9 ± 5.8
Long-life/UHT whole fluid milk, cow’s, branded	20	29	As such	4.4 ± 0.3	4.5 ± 0.4
Chicken breast, bone-in, skin-on, frozen/raw, branded	13	27	Cooked	34.8 ± 3.8	57.5 ± 8.8
Beef chuck eye steak, chilled/raw	13	19	Cooked	32.3 ± 4.4	45.5 ± 2.0
Chicken egg, large size, raw, branded	13	17	Cooked	9.0 ± 0.9	14.1 ± 1.7
Brazilian wheat bread roll (*pão francês*), baked	13	20	As such	9.7 ± 2.3	11.3 ± 2.9
Ground coffee, branded	21	30	Brewed, 8%	1.4 ± 0.1	1.7 ± 0.2

^1^ For non-branded food items (e.g., orange, onion, raw/chilled beef chuck, etc.), the collected regular price was the one available in each visited supermarket/grocery store; for branded food items (e.g., soybean oil, rice, and ground coffee), where more than one brand was frequently available in each supermarket, the collected regular price was always the lowest; “best buys” or limited-time discount offers were not considered. ^2^ For detailed information on food characteristics/composition, forms of consumption, forms of preparations, standard serving size, edible conversion factor, cooking conversion factor, etc., see Appendix A Table A1. ^3^ The number of samples of food staples corresponds to the number of visited supermarkets/grocery stores where the item was found. ^4^ To obtain the average price per g or mL of a given converted food (i.e., food without inedible/unwanted parts and then cooked, if applicable), first the reference weight (in grams) or volume (in mL) of food for sale in supermarkets/grocery stores (see Appendix A Table A1) is transformed into weight or volume of edible/cleaned/cooked food (this is conducted by multiplying the reference weight or volume of food for sale by the total conversion factor, TCF; see Appendix A Table A1 for values). Then, the average price (in Brazilian reals, BRL) of the food per reference weight (in g) or volume (in mL), as collected in supermarkets/grocery stores (not shown), is divided by the weight (g) or volume (mL) of the converted food (e.g., 1000 g of raw beef chuck eye steak, with TCF = 0.5882, converts into 588.2 g of cooked beef prepared/shallow fried with onion, garlic, vegetable oil, and salt; visible fat trimmed before cooking). To obtain the average price of food per reference weight (g) or volume (mL), as sold in supermarkets/grocery stores, multiply the average cost (in BRL) of converted food per g or mL (see table above) by the TCF and by the reference weight (g) or volume (mL) for sale (e.g., the average cost per g of edible/cooked beef chuck eye steak in August–September/2020 is BRL 0.0323; thus, BRL 0.0323/g × 0.5882 × 1000 g = BRL 19.00 per kg or 1000 g of raw/chilled beef chuck eye steak).

**Table 6 nutrients-16-01469-t006:** Daily expenditure (BRL) in 9 scenarios of alcohol per capita consumption in Brazil for moderate (16.8 g pure alcohol/day), risky (41.7 g/g/day), and heavy (83.4 g/day) drinking, considering the consumption of beer only, cachaça only, or a combination of the two beverages (64% beer + 36% cachaça).

Type of Alcoholic Beverage ^1^	Consumption in g of Pure Alcohol/Day ^2^	Consumption in mL of Beverage per Day ^3^	Daily Expenditure (BRL) ^4^	
			2020	2021
Beer only	16.8	452.5	2.52	2.86
	41.7	1123.1	6.26	7.09
	83.4	2246.2	12.51	14.18
Cachaça only	16.8	53.8	0.67	0.67
	41.7	133.6	1.67	1.67
	83.4	267.3	3.34	3.34
Combination (64% beer + 36% cachaça)	10.8 + 6.0	290.9 + 19.2	1.86	2.08
	26.7 + 15.0	719.1 + 48.1	4.61	5.14
	53.4 + 30.0	1438.2 + 96.1	9.21	10.28

^1^ Alcoholic beverages (beer and cachaça) were all low-price leading national brands surveyed in local (João Pessoa, Brazil) supermarkets/grocery stores in August–September/2020 and August–September/2021 (see Table 4 for more details). ^2^ Values (g of pure alcohol/day) for moderate, risky, and heavy drinking were obtained as described in Table 1. ^3^ The corresponding volume (mL) of the beverages (beer or cachaça) was calculated as follows: [(consumption in g of pure alcohol per day)/0.79] × (1/alcohol content of each beverage in %vol). The alcohol contents of beer and cachaça used were 4.7% and 39.5%, respectively, because they correspond to the average alcohol contents displayed on labels of beers and cachaças selected for the present study. The density of alcohol is 0.79 g/mL. Values were calculated as described in Table 2. ^4^ See Table 4 for average prices of beers, cachaças, and of g of pure alcohol of corresponding beverage.

**Table 7 nutrients-16-01469-t007:** Daily expenditure (BRL) on a balanced ^1^ staple diet of 2000 kcal, consisting of 5 meals, and corresponding consumed quantities (g or mL) of each food item and their calories (kcal).

Converted ^2^ Low-Price Food Staples ^3^	Consumption ^4^	kcal ^5^	Expenditure (BRL) ^6^	
			2020	2021
Breakfast (517.2 kcal): various				
Papaya cv Formosa, raw/natural, sliced	60 g	28.2	0.21	0.32
Salted/steamed flaked cornmeal (“Brazilian corn couscous”)	220 g	246.4	0.33	0.37
Salted/shallow fried (soybean oil) chicken egg	48 g	111.8	0.43	0.68
Unsalted butter, spread on “corn couscous”	10 g	77.6	0.50	0.52
Black coffee (via infusion brewing at 8%)	165 mL	13.2	0.23	0.28
White granulated sugar (added to coffee)	10 g	40.0	0.02	0.04
Morning snack (134.0 kcal): milk banana smoothie				
Long-life/UHT whole fluid milk	100 mL	65.0	0.44	0.45
Banana cv. Pacovã, raw/natural, sliced (blended with milk)	60 g	69.0	0.21	0.33
Lunch (594.5 kcal)				
Cooked/boiled beans cv Carioca (prepared with onion, garlic, water, soybean oil, and salt; served with 50% beans and 50% broth)	100 g	79.0	0.25	0.25
Cooked/boiled white rice (prepared with onion, garlic, water, and soybean oil)	250 g	347.5	0.48	0.49
Green butterhead lettuce, raw/natural, sliced	50 g	6.5	0.24	0.41
Carrot, raw/natural, grated	50 g	15.5	0.19	0.23
Cooked/shallow fried beef chuck eye steak (prepared with onion, garlic, soybean oil, and salt; served drained)	50 g	106.0	1.61	2.27
Orange juice, raw/natural, squeezed/strained	120 mL	40.0	0.49	0.59
Afternoon snack (306.0 kcal): cheese sandwich				
Brazilian wheat bread roll (pão francês)	70 g	210.0	0.68	0.79
Mozzarella-type cheese, cow’s, sliced	30 g	96.0	1.08	1.12
Dinner (448.3 kcal): various				
Cooked/boiled spaghetti-type wheat pasta, unsalted, drained	250 g	315.0	0.35	0.45
Cooked/shallow fried chicken breast (prepared with soybean oil and salt; served drained, boned, and skinned)	50 g	93.0	1.74	2.88
Tomato cv. Italiano, raw/natural, sliced	85 g	15.3	0.24	0.30
Pineapple cv. Pérola, raw/natural, sliced	50 g	25.0	0.26	0.27
Approximate quantities of converted food ingredients ^7^ and cooking gas used in all culinary preparations.				
Soybean oil, refined	65 mL	- ^9^	0.40	0.62
White onion, raw	120 g	- ^9^	0.37	0.50
Garlic, raw	57 g	- ^9^	1.75	1.39
Table salt, refined	5 g	- ^9^	0.01	0.01
Cooking gas (liquefied petroleum gas, LPG) ^8^	0.11 kg	-	0.60	0.80
	Total ^10^	2000.0	13.11	16.36

^1^ The above diet is considered balanced because its daily energy and nutrient profile [Energy, 2000 kcal; carbohydrates (starches and sugars), 296.02 g (59.2% energy); estimated free sugars, 48.76 g (9.8% energy); protein, 82.10 g (16.4% energy); total fat, 48.32 g (21.7% energy); saturated fat, 18.71 g (8.4% energy); sodium, 1852.91 mg; fruits and vegetables, 475.0 g; fiber, 26.05 g]—calculated from the energy and nutrient content values in Appendix A Table A1 and Table A2—is in line with levels/recommendations of dietary components presented in Table 3; estimates of free sugars intake were based on the quantities of selected sugar-containing food in the diet and their corresponding free sugar contents (%), namely fruits—papaya, 8%; banana, 15%; orange, 9%; pineapple, 9% [27]; vegetables—carrot, 6%; tomato, 2% [28]; whole milk, 5% [29]; granulated sugar, 99.6% [15]. ^2^ Converted food is the edible food (i.e., without inedible parts) that has been prepared (cooked, if applicable) in a specific way and that is ready to be consumed or used; the food preparation and consumption in this study considered known dietary habits in Brazil and information from the Brazilian Food Composition Table [15]. ^3^ For detailed information on food characteristics/composition, forms of consumption, forms of preparations, standard serving size, edible conversion factor, etc., see Appendix A Table A1. ^4^ Quantities (g or mL) of converted food consumed in each meal (for information on standard serving size of each food item, see Appendix A Table A1). ^5^ Energy (kcal) in the converted food, as shown in Appendix A Table A1. ^6^ The average expenditure on food items, as collected in August–September 2020 (2020) and August–September 2021 (2021), was calculated by multiplying the consumed quantities by the corresponding average price of converted food presented in Table 5. ^7^ The quantity of ingredients used in all culinary preparations was estimated by multiplying the standard serving size of each ingredient (13 mL, soybean oil; 40 g, onion; 19 g, garlic; 1 g, salt; see Appendix A Table A1) by the number of preparations in which they were used, namely 5 (soybean oil), 3 (onion), 3 (garlic), and 5 (salt). ^8^ The following formula was used to calculate the daily per capita expenditure (BRL) on LPG in 2020 and 2021, respectively: (0.11 kg/13 kg) × BRL 71.42 and (0.11 kg/13 kg) × BRL 94.30; where 0.11 kg is the estimated daily LPG demand per capita in Brazilian households and 13 kg is the most common size of LPG bottle used in Brazilian households [30]; BRL 71.42 and BRL 94.30 are the average prices of a 13 kg LPG bottle in the city of João Pessoa, as collected in 19 and 23 LPG retail outlets by the city of João Pessoa Consumer Protection Agency (Procon-JP) in August 2020 and August 2021, respectively [31,32]. ^9^ Individual energy (kcal) and nutrition values of food ingredients used in culinary preparations (soybean oil, onion, garlic, and salt) are not considered for calculations in the diet because they are already included in the energy and nutrient content of the converted food. ^10^ “Total” refers to total energy (kcal) in the diet (sum of calories of each consumed converted food item) and total expenditure (BRL) on the diet (sum of average cost of each consumed item) in 2020 and 2021.

## Data Availability

The raw data supporting the conclusions of this article will be made available by the authors on request.

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
