# Peer review of "The Paradox of Alcohol and Food Affordability: Minimal Impact of Leading Beer and Cachaça Brands on Brazilian Household Income Amid Hazardous Drinking Patterns"

_nutrients, 2024, doi:10.3390/nu16101469_

Round 1

Reviewer 1 Report

Comments and Suggestions for Authors

Dear Authors,

Thank you for this submission. An interesting article.

To improve the paper, please consider the following:

Page 1: line 13 - change 3 million to three million

Page 1: line 21 - 3 scenarios to three scenarios

Keywords: check how many are admissible. 11 seems to many

Page 1: line 34 - Reference 1 - Is there a more recent statistic post covid? 

The materials and methods section is well written and easy to understand.

Page 3: lines 99 - 101. Can you mention if it cost anything to obtain the WHO data or publicly obtained?

Page 6: I would prefer at least two sentences to form a paragraph but this is a personal preference.

As the results are extremely detailed, I am assuming they are correct. Overall, there are presented well and easy to follow.

Page 15: line 537 - Abramson et al. study was conducted in 2006. Although a good seminal study, it is almost 20 years old. Is there any more recent studies factoring inflation?

Page 15: lines 552-596 - This reads like a literature review as you are introducing background information at this late stage. I would consider moving it to set up why this article is important.

Page 16: lines 597 - 624 - I would put this information after the conclusion.

Page 17: lines 632 - 639 - Future research..... I move to after limitations.

Author Response

Dear Authors,

Thank you for this submission. An interesting article.

RESPONSE: Thank you for your suggestions to improve our paper.

To improve the paper, please consider the following:

Page 1: line 13 - change 3 million to three million

RESPONSE: Done.

Page 1: line 21 - 3 scenarios to three scenarios

RESPONSE: Done.

Keywords: check how many are admissible. 11 seems to many

RESPONSE: The keywords were reduced in number to 8.

Page 1: line 34 - Reference 1 - Is there a more recent statistic post covid? 

RESPONSE: No. The most current version of the WHO status report is from 2018.

The materials and methods section is well written and easy to understand.

RESPONSE: Thank you.

Page 3: lines 99 - 101. Can you mention if it cost anything to obtain the WHO data or publicly obtained?

RESPONSE: The data is available for free without any cost. “Publicly available” was added as requested.

Page 6: I would prefer at least two sentences to form a paragraph but this is a personal preference.

RESPONSE: Thank you for the suggestion. The number of paragraphs was reduced on page 6.

As the results are extremely detailed, I am assuming they are correct. Overall, there are presented well and easy to follow.

RESPONSE: Thank you for your positive feedback on the clarity and presentation of the detailed results in our manuscript. We appreciate your assumption of their accuracy and have made every effort to ensure that all data are thoroughly validated and accurately reported. We are glad to hear that the results are easy to follow, as we aimed to present complex data in a comprehensible manner.

Page 15: line 537 - Abramson et al. study was conducted in 2006. Although a good seminal study, it is almost 20 years old. Is there any more recent studies factoring inflation?

RESPONSE: Thank you for your insightful comment. You correctly point out the age of the Abramson et al. study and the importance of considering more recent economic factors such as inflation. While there is no newer research paper than Abramson et al. available, our study acknowledges the necessity of updating these findings with current economic conditions. To address this, we have incorporated the latest available data on food and alcohol prices from 2020 and 2021 to reflect current market conditions in Brazil. These recent data points allow us to evaluate the impact of inflation and other economic changes that have occurred since the Abramson et al. study. This approach not only updates the seminal findings but also enhances the relevance and applicability of our research to contemporary policy-making and economic realities.

Page 15: lines 552-596 - This reads like a literature review as you are introducing background information at this late stage. I would consider moving it to set up why this article is important.

RESPONSE: Thank you for the suggestion. We have moved part of the material to the introduction.

Page 16: lines 597 - 624 - I would put this information after the conclusion.

RESPONSE: We agree that the flow was not optimal in our text. The paragraph was moved to the end of the discussion.

Page 17: lines 632 - 639 - Future research..... I move to after limitations.

RESPONSE: The “future research” was moved to the very end of the conclusion as requested.

Reviewer 2 Report

Comments and Suggestions for Authors Alcohol consumption, associated with various cancers, mental disorders, and aggressive behaviour, leads to mass deaths globally each year. The WHO aims to reduce alcohol consumption, emphasizing the need for tailred national policies to address this issue. The manuscript (nutrients-2983965-peer-review-v1) submitted by Ian C.C. Nobrega et al. describes the costs of consuming leading brands of beer and sugarcane spirit cachaca against the expenditure on staple foods: The paradox of alcohol and food affordability: minimal impact of leading beer and cachaca brands on Brazilian household income amid hazardous drinking patterns. Results showed that all heavy consumption scenarios cost less or significantly less than 2000 kcal/day staple diet, highlighting an urgent need for fiscal policies, such as a minimum unit pricing for alcohol, to address public health concerns. The content of this MS is abundant. This MS needs revision before publication. See my comments below to improve this manuscript. 1. The social factors also need to be considered in the manuscript. 2. The dietary habits is an important factor to alcohol consumption. 3. The correlation between income and education is also an important factor to alcohol consumption, and need to be considered in the study.

Author Response

Alcohol consumption, associated with various cancers, mental disorders, and aggressive behaviour, leads to mass deaths globally each year. The WHO aims to reduce alcohol consumption, emphasizing the need for tailred national policies to address this issue. The manuscript (nutrients-2983965-peer-review-v1) submitted by Ian C.C. Nobrega et al. describes the costs of consuming leading brands of beer and sugarcane spirit cachaca against the expenditure on staple foods: The paradox of alcohol and food affordability: minimal impact of leading beer and cachaca brands on Brazilian household income amid hazardous drinking patterns. Results showed that all heavy consumption scenarios cost less or significantly less than 2000 kcal/day staple diet, highlighting an urgent need for fiscal policies, such as a minimum unit pricing for alcohol, to address public health concerns. The content of this MS is abundant. This MS needs revision before publication. See my comments below to improve this manuscript.

RESPONSE: Thank you for your assessment of our paper!

  1. The social factors also need to be considered in the manuscript.

RESPONSE: The social factors were not part of the investigations. However, we have expanded the part about social factors in the limitations section of the discussion.

  1. The dietary habits is an important factor to alcohol consumption.

RESPONSE: Thank you for raising this point. While the interaction between dietary patterns and alcohol consumption was not part of the study investigation, we have now included this important point in the discussion section, along with the other socioeconomic factors.

  1. The correlation between income and education is also an important factor to alcohol consumption, and need to be considered in the study.

RESPONSE: The intersection of income and education clearly creates a complex socio-economic landscape that can dictate the accessibility, choices, and consumption habits of foods and alcohol. Nevertheless, education was not part of our investigation and research question. We now, however, stress the importance of education among the socioeconomic determinants at the beginning of the discussion section.

Round 2

Reviewer 2 Report

Comments and Suggestions for Authors

The manuscript did not take into account the interaction of other factors. The conclusion of the study is unfair.

Author Response

The manuscript did not take into account the interaction of other factors. The conclusion of the study is unfair.

RESPONSE: Thank you for your continued feedback and critical assessment of our manuscript.

We acknowledge your concern regarding the exclusion of certain interactions of factors, such as social, dietary, and educational influences on alcohol consumption. While these factors are indeed significant, our study was primarily designed to explore the economic aspects of alcohol consumption, specifically the impact of alcohol pricing on household income and expenditure on staple foods.

In response to the previous comments in peer review round #1, we have carefully reviewed our submission and now highlight sections within the text where the factors in question were mentioned. We recognize the importance of these factors; however, our study was specifically designed with a focus on the economic implications of alcohol consumption in relation to household income and expenditure on staple foods in Brazil. The study methodology does not allow to statistically assess interactions with other factors.

Given the specialized nature of our study question, expanding the scope to include extensive analysis on social, dietary, and educational factors would divert from our original research objectives and exceed the data and resources we have utilized. As such, extensive new research into these areas is not feasible within the constraints of our current study framework.

We believe the highlighted sections in the manuscript effectively acknowledge these factors within the context of our study's focus, and we remain committed to our research's contribution to understanding the economic aspects of alcohol consumption. We appreciate your understanding.

Finally, we would like to emphasize that our submission was intended as a communication, not as a comprehensive research article. This format was specifically chosen to succinctly address the focused topic of economic implications of alcohol consumption relative to household expenditures on food in Brazil. Communications are typically concise and targeted in scope, intended to quickly disseminate important findings or novel insights. Given this context, the depth and breadth of the study are aligned with the expectations for a communication, where the exploration of broad social, educational, and dietary factors might not be as detailed as it would be in a full-length article.

Considering the request to make the conclusion “fairer”, we aimed to moderate the language to acknowledge the potential of proposed policies rather than suggesting certainty about their outcomes. This approach reflects a balanced perspective, considering both the effectiveness and the limitations of such measures.